# Autonomous Sensory Meridian Response (ASMR): a flow-like mental state

Emma L. Barratt and Nick J. Davis

Department of Psychology, Swansea University, Swansea, United Kingdom

## ABSTRACT

Autonomous Sensory Meridian Response (ASMR) is a previously unstudied sensory phenomenon, in which individuals experience a tingling, static-like sensation across the scalp, back of the neck and at times further areas in response to specific triggering audio and visual stimuli. This sensation is widely reported to be accompanied by feelings of relaxation and well-being. The current study identifies several common triggers used to achieve ASMR, including whispering, personal attention, crisp sounds and slow movements. Data obtained also illustrates temporary improvements in symptoms of depression and chronic pain in those who engage in ASMR. A high prevalence of synaesthesia (5.9%) within the sample suggests a possible link between ASMR and synaesthesia, similar to that of misophonia. Links between number of effective triggers and heightened flow state suggest that flow may be necessary to achieve sensations associated with ASMR.

## INTRODUCTION

In recent years, there has been growing interest in a previously unknown sensory phenomenon, named Autonomous Sensory Meridian Response (ASMR) by those capable of experiencing it. Those who describe ASMR claim it to be an anomalous sensory experience which has thus far escaped the eye of scientific research. There is a suggestion that ASMR may be of use for providing temporary relief to individuals with depression, stress and chronic pain. As ASMR has received some media attention in recent months, many have taken to public forums to explain their ability to induce ASMR to ease symptoms of these conditions in cases where other routes of treatment may have been lacking or ineffective (*Taylor, 2013*; *TheWaterwhispers, 2013*), while others use ASMR exclusively as a relaxation tool (*Marsden, 2012*). To date there has been no rigorous scientific exploration of ASMR, nor of the conditions which trigger or end the ASMR state.

Media designed specifically to produce ASMR has amassed a community of thousands of members. Capable individuals utilise a variety of visual and audio stimulation—most typically through video sharing—to achieve a tingling, static-like sensation widely reported to spread across the skull and down the back of the neck (*Taylor, 2014*). The advent of online video communities has facilitated a gathering of those who experience ASMR, and as a result hundreds of videos have been produced, viewed and shared with

Corresponding author
Nick J. Davis,
n.j.davis@swansea.ac.uk

**Table 1 Popular ASMRtists.** Popular ASMR-related channels on YouTube. Counts correct as of 10 December 2014.

| Name | Channel URL | Total views |
|---|---|---|
| WhisperTalkStudios | https://www.youtube.com/user/WhisperTalkStudios | 218,900 |
| GentleWhispering | https://www.youtube.com/user/GentleWhispering | 88,311,107 |
| MassageASMR | https://www.youtube.com/user/MassageASMR | 46,575,761 |
| Fairy Char ASMR | https://www.youtube.com/user/feirychaRstaRs | 9,008,828 |
| Ephemeral Rift | https://www.youtube.com/user/EphemeralRift | 27,053,163 |
| ASMRRequests | https://www.youtube.com/user/ASMRrequests | 648,590 |
| TheUKASMR | https://www.youtube.com/user/TheUKASMR | 7,734,238 |

the goal of inducing this sensation, which is said to be paired with a feeling of intense relaxation. A dedicated ASMR subgroup on Reddit (http://www.reddit.com/r/asmr/) boasts 86,000 subscribers from around the world, and some of the most popular ASMR content creators on video sharing site Youtube (http://youtube.com/), for example GentleWhispering have upwards of 300,000 subscribers. Table 1 lists a number of these popular sources on Youtube. These figures show that the culture surrounding ASMR is in no way insignificant. Several reputable international media outlets have reported on the attention this phenomenon is receiving, and the lack of scientific explanation. (*Marsden, 2012*; *Tomchak, 2014*).

Though stimuli used to induce ASMR are widely varied, and devotees report that individual differences play a pivotal role in the effectiveness of each video, distinct themes appear to be present in ASMR media. Exploration of the most viewed ASMR media on Youtube uncovers what may be discrete categories of common triggers. For example, many of these videos depict role play situations, in which the viewer is placed in a position of 'close proximity' to another person in order to be cared for in some manner. Often this involves grooming (e.g., MassageASMR; Fairy Char ASMR), or being given some type of medical examination (e.g., WhisperTalkStudios). The tone of these types of ASMR media is usually one of having close attention paid to you, the viewer, with videos shot in a point of view manner. Other videos include acts which require a similar amount of focus, but directed towards objects, rather than the viewer (e.g., Ephemeral Rift).

ASMR videos also typically appear to include an emphasis on the use of sound to trigger the static sensation of ASMR, which include the subjects of these videos cycling through a variety of household items which make various noises when tapped upon or used (e.g., MassageASMR). On the surface, this trigger resulting in sensation seems quite similar to the experience of synaesthesia, a phenomenon in which specific external stimuli cause an internal experience in a second, unstimulated modality (*Banissy, Jonas & Kadosh, 2014*). The reported automatic, consistent response to audio-visual stimuli which is felt in tactile sensory modalities alongside a feeling of calm does appear to resemble synaesthesia in these aspects, though the tactile concurrents (secondary sensations in the unstimulated modality; *Cytowic, 2002*) found in ASMR appear to be more tangible than those experienced in synaesthesia (ie. tingling on the skin). Even with this being the case,

**Peer**J

the positive emotional response of calm said to be triggered by ASMR media consumption could potentially be considered a form of sound-emotion synaesthesia.

Reports of ASMR experiences also appear to share some features with the state of "flow," which is the state of intense focus and diminished awareness of the passage of time that is often associated with optimal performance in several activities, including sport (*Csikszentmihalyi & Csikzentmihaly, 1991*; *Swann et al., 2014*). Anecdotal reports of ASMR describe states of focus, of greater "presence" and of relaxation which are consistent with the non-active aspects of flow.

The aim of the current study was to describe the sensations associated with ASMR, explore the ways in which it is typically induced in capable individuals, and to provide further thoughts on where this sensation may fit into current knowledge on atypical perceptual experiences. This research also aims to explore the extent to which engagement with ASMR may ease symptoms of depression and chronic pain. As ASMR has yet to be defined within scientific literature, this study will utilise survey data and qualitative descriptive contributions from participants to explore the characteristics of ASMR, and to provide a basis for later experimental investigation.

## MATERIALS AND METHODS

### Participants

The sample of the present study was comprised of 245 men, 222 women and 8 individuals of non-binary gender ($N = 475$). These participants presented themselves as volunteers via online advertisement on specialised ASMR interest groups on Facebook and Reddit. The age of the sample ranged from 18 to 54 years (mean $= 24.6$ years, st. dev. $= 7$ years). Volunteers were located worldwide, with particular participation from the United States of America and Western Europe. All individuals in the sample self-reported to have experienced ASMR and regularly consume ASMR media.

### Method

An online questionnaire (www.qualtrics.com, Version 36,892) was conducted in order to gather information on the prevalence of particular features of ASMR, when and why individuals engage in ASMR, and the relation of ASMR to other known phenomenon. Ethical approval was granted by the Department of Psychology of Swansea University, and continuation from the initial screen of this questionnaire, which contained a brief summary of the research topic and all necessary ethical information, served as informed consent. The structure of this questionnaire is described below, and a version of the text of the questionnaire is included as Appendix S1:

#### Section 1—demographics

Demographic information, including whether or not individuals suffered from any chronic illness or took medications, was gathered at the beginning of the survey. In addition, the Beck Depression Inventory (BDI-II; *Beck, Steer & Brown, 1996*) and Beck Anxiety Inventory (BAI; *Beck et al., 1988*) were included to give insight on the daily mood of participants. As several online sources indicate the existence of a subset of ASMR media

users who engage in ASMR to manage symptoms of depression, stress, or pain, this data would be used to explore efficacy of ASMR in easing symptoms of these conditions. Participants were asked to verify that they identified as able to experience ASMR and the tingling sensations commonly associated with ASMR. No leading elaboration was given with regard to this sensation, as all participants had been recruited via ASMR social network groups, and would therefore be aware of how this aspect of the phenomenon is typically described. This was an attempt to limit imposing researcher assumptions about ASMR. In this section, participants were also given a definition of synaesthesia, alongside some examples of synaesthetic associations. Participants were asked to report if they suspected they may experience any type of synaesthesia. Those who responded in a positive or unsure manner were asked to specify which type of synaesthesia they thought they may have, and were followed up approximately four weeks later via e-mail to be assessed for consistency.

### Section 2—viewing habits

This section included questions pertaining to how often participants engaged in ASMR media sessions, how many videos they consumed in a single session, and at what time of day they typically viewed ASMR media. Questions regarding the optimal conditions to experience ASMR were also included.

### Section 3—triggers

Participants were asked to report whether or not they experienced any of the triggers in a list of 9 given stimuli: Crisp sounds, whispering, personal attention, vacuum noise, aeroplane noise, laughing, smiling, watching repetitive tasks, and slow movements. Of these suggestions, five possible triggers were inspired by the typical content of ASMR videos (e.g., Close personal attention, crisp sounds) and four were unlikely triggers (vacuum noise, aeroplane noise, laughing, smiling). These unlikely triggers are commonly present in ASMR videos, but are not commonly identified in titles or online discussions, so were considered to be unlikely to produce tingles in many participants. This section included a comment box in which participants could specify what, if anything, abolished the tingling sensations. Preference of receiving auditory triggers in one ear over another was also probed.

### Section 4—location

In order to more clearly define the location and time course of the tingling sensation associated with ASMR, participants were asked to report where on their body they typically felt tingles originate, and whether or not the sensation always originated in that area. Participants were also asked whether or not the tingling evolved or spread with intensity, and if so, which other body areas the tingling sensation spread to.

### Section 5—'Why do you watch ASMR videos?'

This section presented several likert style statements to be rated from 'strongly agree' to 'strongly disagree' in terms of how well each represented individuals' experiences of ASMR and ASMR media. These included statements concerning mood and arousal control, such

as 'I watch ASMR videos to relieve negative mood,' '... to deal with anxiety,' and '... to relieve stress.' Further, more generalised statements, such as 'I know what triggers my ASMR,' 'I watch ASMR videos for sexual stimulation,' and 'ASMR videos help me focus' were included to obtain a rounded view of why participants choose to engage with ASMR media.

### Section 6—flow state scale

Since the reported ASMR experience shares some features with that of the 'flow' state (*Csikszentmihalyi & Csikzentmihaly, 1991*), we used a reduced version of the Flow State Scale (*Jackson & Marsh , 1996*) to quantify this experience. We selected only the eight questions relating to the passive experience of flow. Participants scored their agreement with statements such as "Things seem to happen automatically" on a 5-point scale. These scores were initially subjected to factor analysis to confirm that only a single factor had been captured in the reduced questionnaire. Combined scores, composed of the sum of the scores of the components, were then submitted to Pearson's Correlation to investigate links between flow state and trigger thresholds.

### Section 7—Effect on mood and chronic pain

Using an interactive sliding scale ranging from 0 to 100, participants were asked to rate their experience of mood during a typical day, directly before, during, one hour after and 3 h after a successful ASMR media viewing session. These ratings were given at the same time, i.e., not during the ASMR state. 0 on this scale represented 'terrible, the worst I've ever felt,' whereas 100 represented 'euphoric, the best I've ever felt.' Participants who earlier indicated that they suffered from chronic pain were also asked to complete a version of this task with the intensity of their pain symptoms in mind.

## Data analysis

Where possible, analyses were conducted on the entire sample ($N = 475$). However, due to certain sections being inapplicable to some participants, some sections included data from a subset of the entire sample. In these cases, $N$ is reported alongside the results. All analyses were carried out in SPSS and Microsoft Excel. A copy of the data from this experiment are included as Data S1.

## RESULTS

### Why engage in ASMR?

Through Likert style questions, participants largely sought out ASMR as an opportunity for relaxation, with 98% of individuals agreeing, or agreeing strongly with this statement. In a similar vein, 82% agreed that they used ASMR to help them sleep, and 70% used ASMR to deal with stress. A small number of individuals (5%) reported using ASMR media for sexual stimulation, with the vast majority of participants (84%) disagreeing with this notion.

Many participants described additional details of seeking the effects of ASMR where other interventions, medical or otherwise, had been unable to assist. This is perhaps best illustrated by a correspondence from one participant whose anxiety and stress was causing significant issues in his daily functioning. After noticing during a hairdressing

**Table 2 Common triggers.** Percentage of participants that reported induction of tingling sensations from each trigger type.

| Trigger type | Percentage of participants triggered |
|---|---|
| Whispering | 75% |
| Personal attention | 69% |
| Crisp sounds (metallic foil, tapping fingernails, etc.) | 64% |
| Slow movements | 53% |
| Repetitive movements | 36% |
| Smiling | 13% |
| Aeroplane noise | 3% |
| Vacuum cleaner noise | 2% |
| Laughing | 2% |

appointment that he felt at ease, he sought out ways to replicate this feeling daily in order to manage his symptoms, and in the process discovered ASMR media. In his own words:

"*I was totally amazed, I can only describe what I started feeling as an extremely relaxed trance like state, that I didn't want to end, a little like how I have read perfect meditation should be but I never ever achieved.*"

## Common triggers

Analysis of responses found four prominent categories of triggers, each experienced by over 50% of participants. These triggers are whispering (75%), personal attention (69%), crisp sounds (64%) and slow movements (53%). 34% of participants also reported that their ASMR was triggered by watching repetitive tasks. Triggers less commonly associated with ASMR media (smiling, vacuum cleaner noise, aeroplane noise, and laughing) were included for comparison. Each of these non-triggers were in each case reported to be effective by less than 3% of participants. These values are illustrated in Table 2. Some individuals reported only being triggered by new viewing material, in which they are unable to predict which trigger will be presented next.

The most common time for engagement with ASMR media was reported to be before going to sleep at night, with 81% of participants reporting this as their preferred time. 4% of participants engaged in ASMR upon waking, 2% participated during the morning to midday. 30% of participants also reported viewing ASMR media in their spare time, regardless of the time of day.

When asked if participants preferred any specific environmental conditions for viewing, 52% responded 'yes.' Submitted comments suggested that of these, individuals near universally preferred quiet, relaxed conditions in order to achieve ASMR from online media. Many also specified preference for binaural headphones, so as to experience depth of sound.

Most participants reported having their first experience of ASMR at age five (65 individuals), with the vast majority (241 individuals) reporting the first experience of ASMR between five and ten years of age. There were also several instances of ASMR being first experienced further into adulthood—41 individuals reported their first ASMR experience as happening after age 18.

## Experience of ASMR

Participants widely reported sensations similar to that found in general reporting of ASMR; a tingling sensation which originated typically towards the back of the scalp and progressed down the line of the spine and, in some cases, out towards the shoulders. Many participants also felt that their lower back, arms and legs experienced the sensation, though the amount of area the tingles covered seemed to be determined by the extent to which individuals had been triggered.

Sixty-three percent of participants reported the tingling sensation associated with ASMR to originate consistently in one part of their body, while 27% said this origin varied. Of those that reported a consistent origin, the static tingling sensation was reported to typically originate on the back of the head (41%) and shoulders (29%). When intense, this sensation is able to extend down the line of the spine (50%), arms (25%) and legs (21%), though this does not occur in every session, and every individual does not experience the same route. An illustration of the most common path of these sensations is provided in Fig. 1.

## Medications which affect ASMR

Of the sample who reported taking medication, only three participants responded positively when asked if they had noted effects of any of their medications on ASMR. One participant noted that their antidepressant stifled sensations of ASMR, which later returned once they stopped taking the medication, though they did not specify which. Another noted that sleeping pills dulled their ASMR experience. A third reported that Clonazepam decreased the sensations associated with ASMR. Six participants responded that their medication had no effect on ASMR. One hundred and three other participants who use medication were unsure as to the effect of their medication on their experience of ASMR.

## Effect on mood

Eighty percent of participants responded positively when asked if ASMR has an effect of their mood, while 14% were unsure and 6% felt that ASMR did not alter their mood. When submitted to a mixed ANOVA with factors for time (before, during, immediately following and 3 h after ASMR) and for depression status (high, medium or low as defined by the BDI), we found a significant main effect of time on mood $[F(3.06, 1143.0), p < 0.0005]$. Pairwise comparisons revealed significant differences between all timeframes ($p < 0.0005$ in all cases). Participants reportedly felt best while they are engaging with ASMR media, with reports on the 0 to 100 scale of positive mood averaging at 78 for this time period. The effect on mood steadily decreased over the course of several hours. Means for all time frames are reported in Fig. 2. This effect is moderated by severity of depression, with people at higher risk of depression showing a more rapid decline in mood score over time $[F(10, 2360) = 20.217, p < 0.0005]$ however, there was also a correlation between BDI scores and the difference in mood score between baseline and immediately after an ASMR experience, suggesting that people with higher depression scores had the greatest benefit from engaging in ASMR $[r = 0.439, p < 0.0005]$.

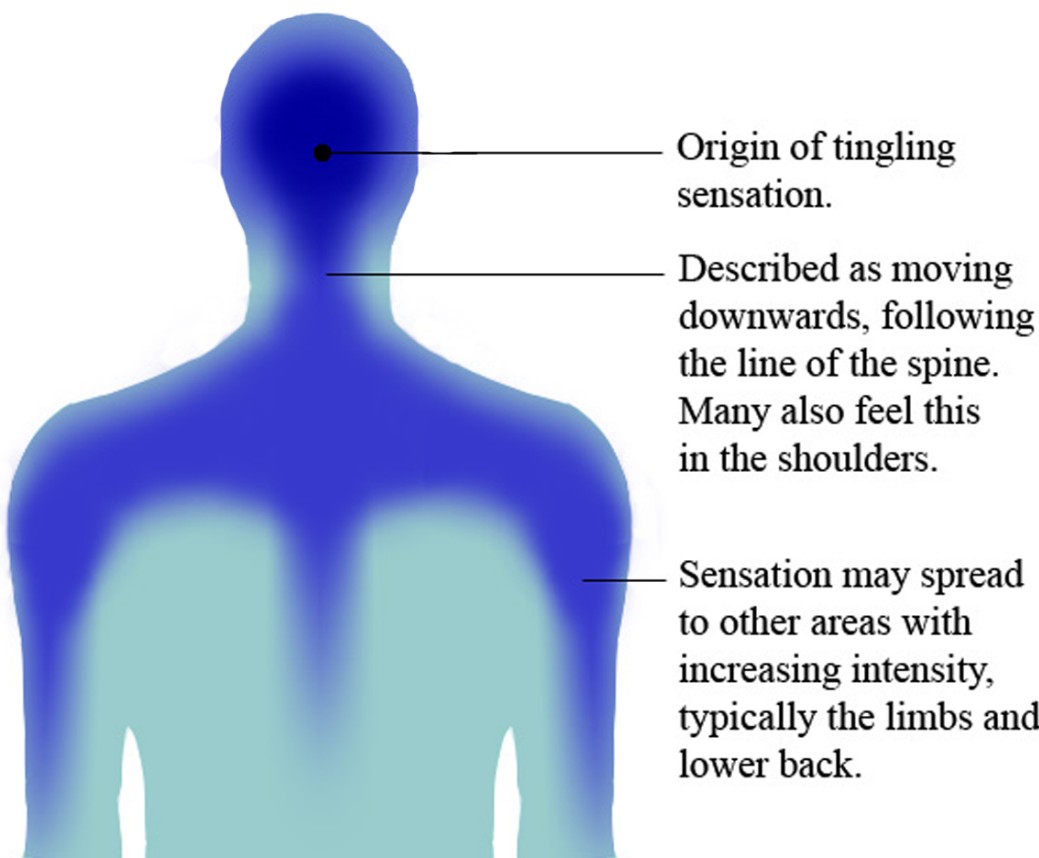

**Origin of tingling sensation.**

**Described as moving downwards, following the line of the spine. Many also feel this in the shoulders.**

**Sensation may spread to other areas with increasing intensity, typically the limbs and lower back.**

**Figure 1 ASMR Map.** An illustration of the route of ASMR's tingling sensation. Image shows rear view of the head and upper torso. Capable individuals typically experience the sensation as originating at the back of the head, spreading across the scalp and down the back of the neck. Half of participants reported that this sensation typically spreads to the shoulders and back with increasing intensity. Though this diagram represents the most common areas involved in the tingling sensation, there is a huge amount of individual variation in where tingles spread to with increased intensity, with legs and arms also commonly reported as hotspots in some individuals.

Fifty percent of participants said their mood improved even in sessions when no tingling sensation was produced, while 30% said that achieving this sensation was vital to mood improvement.

Sixty-nine percent of those who scored moderate to severe on the BDI reported using ASMR to ease their symptoms of depression ($N = 70$). Those scoring as depressed reported a mean improvement in mood of 38.75 (STD $= 18.85$), in comparison to a mean improvement of 21.33 (STD $= 13.58$) in non-depressed participants.

## Effect on chronic pain

Thirty-eight individuals with chronic pain reported that ASMR improved their symptoms. 13 were unsure of ASMR's impact on their symptoms. Forty did not believe that ASMR had an impact on their symptoms of chronic pain. Analyses were carried out on the responses of individuals who responded positively and unsurely to this section. Six individuals who

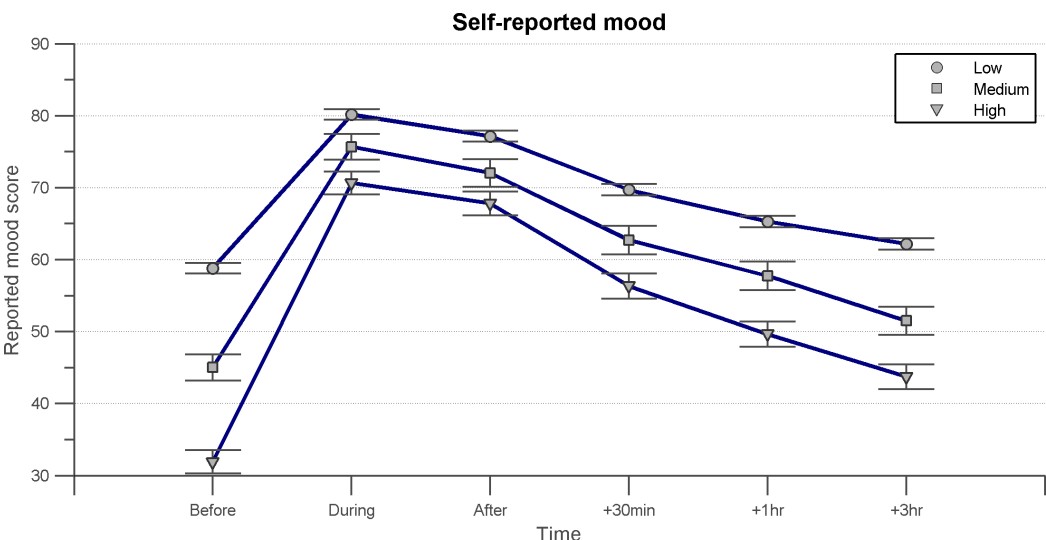

**Figure 2 BDI graph.** The time course of mood before, during, immediately following, and several hours after engaging in ASMR. Data shown is the mean mood score given to each time frame by all participants ($N = 475$), with participants grouped according to their Beck Depression Index. Mood scores could range from 0 to 100, 0 representing the worst the individual had ever felt, 100 representing the best they have ever felt. Error bars represent $\pm 1$ standard error.

originally reported issues with chronic pain were omitted due to incomplete data. Data analysis was therefore carried out on 45 cases.

Self-report data for before, during, immediately after and 3 h after ASMR were analysed using a one way ANOVA, and were found to significantly differ [$F(3, 132) = 13.892$, $p < 0.0005)$]. Pairwise comparisons revealed there to be a significant difference in chronic pain symptoms before and during ASMR ($p < 0.0005$), a difference which was maintained three hours following ASMR ($p = 0.014$). There was no significant difference between symptoms of chronic pain during and immediately after ASMR ($p = 1.00$), nor was there a difference between during and 3 h after ASMR ($p = 0.21$).

## Flow state

Fifty cases did not have complete data for the flow state questionnaire, so were removed from analysis. We were interested in whether people who experience the flow state more readily also experience the ASMR state more readily. To examine this we took the sum of each participant's responses on the flow state questionnaire and correlated this with the total number of ASMR triggers each person reported, from the list of commonly-reported triggers (i.e., whispering, crisp sounds, personal attention, repetitive actions, slow movements, smiling, water pouring). We used a non-parametric Spearman's test, as the trigger data tended to fall into a small number of values. We found a highly significant relationship between flow experience and number of triggers, with greater flow experience being associated with a larger number of triggers [rho $= 0.936$, $p < 0.01$]. This relationship is shown in Fig. 3.

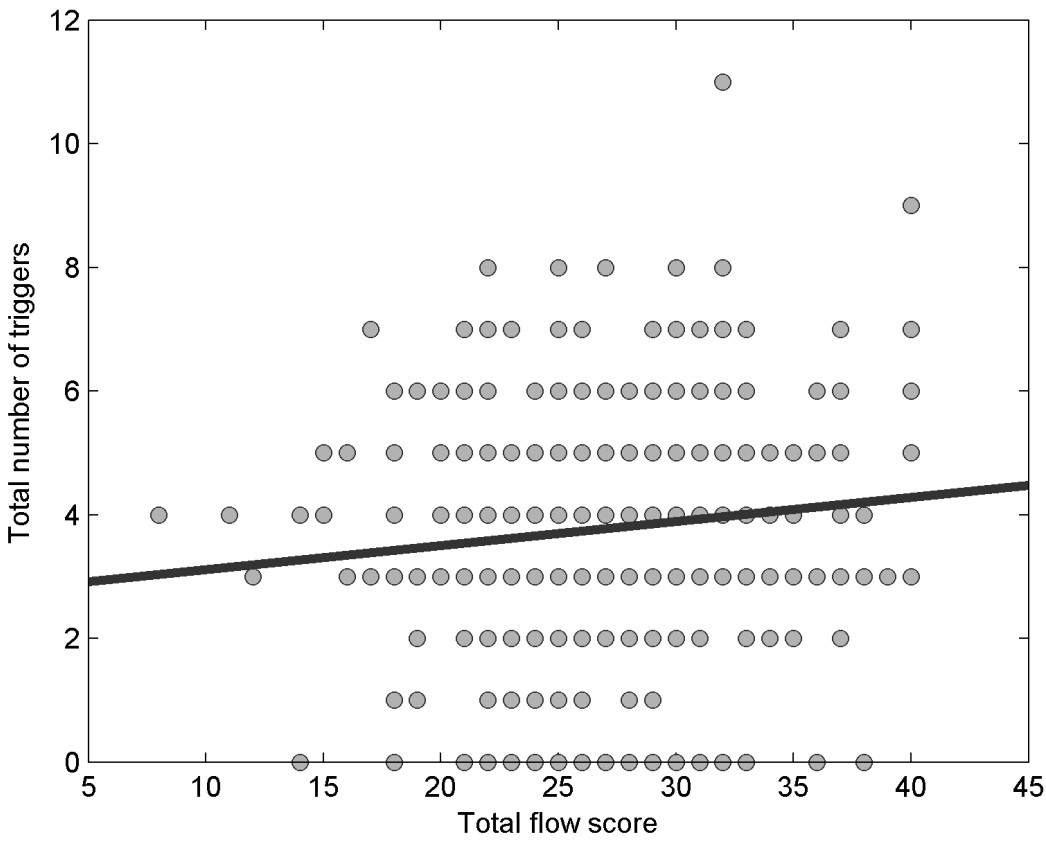

**Figure 3 Flow and Triggers figure.** Relationship between participants' susceptibility to the flow state (expressed as a sum of the scores on the modified Flow State Scale) and the number of triggers of the ASMR state.

## Familial links

When asked if they knew of any family members who experienced ASMR, 38 participants responded positively, 59 responded negatively, and the remaining 378 were unsure or had not inquired. The relations most often identified as experiencing ASMR were sisters (17 individuals), mothers (11), brothers (7) and fathers (4). There were also reports of grandparents experiencing ASMR, though as relational distance increased fewer individuals were reported to be known as able to experience ASMR. It is likely that the perceived strangeness and stigma many individuals feel surrounds ASMR, has prevented many from asking if other individuals within their family experience something similar. The reports gathered through this research, however, do appear to indicate a familial aspect to the ability to experience ASMR.

## Synaesthesia

Synaesthesia appeared to be particularly prevalent within the sample. Thirty-five participants reported experiencing various subtypes synaesthesia and, after exploration of the consistency of concurrents through a follow up interview, 29 of these cases were assessed to be genuine. This places the prevalence of synaesthesia within the sample at

5.9%, compared the current estimate of prevalence in the general population of 4.4% (*Simner et al., 2006*), however, our value fell slightly short of being significantly higher than the general population ($Z = 1.594$, $p = 0.0555$). Participants reported several subtypes, including grapheme-colour, grapheme-personality, time-space and pain-gustatory synaesthesia.

Some comments submitted seem to resemble the inducer-concurrent relationship in synaesthesia. One individual described the tingling sensation as changeable depending on the gender of the voice in the ASMR video she was currently watching. She reported that a female voice would cause the tingles to extend more strongly down one leg, whereas a male voice would increase the sensation in the other leg. Several individuals responded similarly, specifying that '*different triggers hit different parts,*' However, without more data it is difficult to ascertain whether similar experiences are common amongst ASMR capable individuals.

## DISCUSSION

ASMR can be defined as a combination of positive feelings, relaxation and a distinct, static-like tingling sensation on the skin. This sensation typically originates on the scalp in response to a trigger, travelling down the spine, and can spread to the back, arms and legs as intensity increases. An increase in intensity can be achieved through experiencing further triggers.

Those who are able to can engage in ASMR through specialised media at any time, given that the environment in which they attempt to do so is quiet and calm. Many report being triggered by viewing others engaged in focused, precise tasks, by having close personal attention paid to them, or by any number of audio stimuli, such as whispering, tapping or other crisp sounds. Though the effectiveness of various triggers is subject to individual differences, most who experience ASMR can be induced by the above categories of stimuli, either through watching specially designed media, or by coming across triggers in daily life. In capable individuals, ASMR is used mainly to achieve relaxation and for stress relief purposes.

### Uplifting mood and pain relief

The results of this study suggest that ASMR also provides temporary relief in mood for those suffering from depression, with many individuals consciously using it for this purpose. Individuals whose scores on the BDI suggested moderate to severe depression reported a significantly more uplifting effect of engaging in ASMR than those without depression. Those suffering from symptoms of chronic pain also benefitted from ASMR, seeing a significant reduction in their discomfort for several hours following an ASMR session.

Many reported that even in the absence of tingling sensations, they felt that their mood and symptoms of pain had been improved. It is possible that devoting specific time to engaging in ASMR, watching relaxed scenes play out and sitting quietly could be considered a form of mindfulness (*Langer, 1989*). Those who engage in ASMR take time to focus on positive emotions triggered by these stimuli, focusing exclusively on this the task

at hand. This behaviour is very reminiscent of mindfulness practices, which have already been shown by several studies to have positive effect on both conditions (*Kabat-Zinn, Lipworth & Burney, 1985*; *Segal, Williams & Teasdale, 2012*). This categorisation of ASMR as an exercise in mindfulness meditation perhaps best explains the improvements in mood observed in both depressed and non-depressed participants in this study.

## Obtaining flow state

Individuals who scored highly on flow measures reported regularly experiencing a higher number of triggers. This suggests that those who are able to more readily experience flow state during ASMR media consumption are susceptible to more frequent ASMR experiences during their sessions.

Many ASMR videos show individuals in highly focused states (e.g., performing medical exams) or engaged in repetitive tasks (e.g., folding towels). The behaviour of performers during these types of videos often resembles that of someone in flow state—confidently and accurately executing precise tasks. It may be that ASMR is brought about by obtaining a flow-like state, which is in part facilitated by witnessing others in such a state. Similar transference of state from performers to audience have been observed in studies probing the role of mirror neurons (*Rizzolatti, Sinigaglia & Anderson, 2008*). Higher levels of flow may in turn facilitate triggers to be obtained, as could be indicated by results of this study.

## Links with synaesthesia

The prevalence of synaesthesia of any type within the current study's sample was 5.9%, which is high for the estimated prevalence of 4.4% in the general population (*Simner & Hubbard, 2013*). Although the figure reported here did not exceed the estimated level to a statistically significant degree, we would suggest there may be a relationship between the two phenomena. In emotional subtypes of synaesthesia, individuals feel moved to various emotions by inducing stimuli which should, in theory, have no emotional effect on them (e.g., tactile-emotion synaesthesia; *Ramachandran & Brang, 2008*). This sounds strikingly similar to the experience of emotion in ASMR, where emotionally neutral sounds such as tapping and paper tearing, or visual stimuli such as tasks requiring close concentration, bring about a consistent relaxing, stress relieving, positive emotional response.

It is, however, also worth exploring whether or not the experience of ASMR ends with automatic positive emotional reactions to neutral audio and visual stimuli. There may also be merit in exploring automatic negative emotional reactions to external stimuli, and assessing any relation of such an experience to ASMR. Within literature surrounding synaesthesia, a related phenomenon that fits this description does exist, and is known as misophonia. Those who experience misophonia (literally 'hatred of sound') have automatic negative emotional reactions to particular sounds—the opposite of what can be observed in reactions to specific audio stimuli in ASMR. For instance, sufferers report that noises made by humans, such as 'loud breathing or nose sounds' of any volume can produce feelings of disgust, anger, or hatred in a manner which cannot be explained by previously learned associations. (*Schröder, Vulink & Denys, 2013*) Though this condition has not yet been included in the Diagnostic and Statistical Manual (DSM), there has been

movement for misophonia to be recognised as a psychiatric disorder in future revisions, and links between this phenomenon and other perceptual atypicalities such as synaesthesia have been found (*Edelstein, Brang & Ramachandran, 2012*).

There are distinct similarities between the experience of ASMR and Misophonia. In both phenomena, triggering sounds originate from human movements and behaviours. Reactions to these stimuli automatic in both cases, unexplained by previously learned associations, and have some consistency (with the possible exception of some individuals becoming habituated to triggers from ASMR media they have previously viewed). The present study suggests that ASMR, similarly to misophonia, may have a relationship with synaesthesia. Indeed, both experiences seem to follow somewhat synaesthetic patterns; particular inducers (external stimuli, such as whispering, close attention, etc.) produce concurrents (internal perceptual/sensational experiences—in the case of ASMR, tingling and relaxation) in a somewhat predictable manner. It may be the case that ASMR and misophonia are two ends of the same spectrum of synaesthesia-like emotional responses. Whether this hypothetical spectrum, or indeed ASMR alone, can be classified as a type of sound-emotion synaesthesia is however, debateable.

The main issue with relating ASMR to synaesthesia is that, from the data collected here, there does appear to be a difference between the two in terms of tangibility of concurrents. Whereas synaesthetic concurrents are described as 'having a knowledge or sensation of a certain concurrent' (*Simner & Hubbard, 2013*), the tingling sensation associated with ASMR is described in a very physical sense. If we were to consider the concurrent of ASMR as a tingling sensation, as described by participants of the current study, we could with near certainty say that ASMR is not a subtype of synaesthesia. However, this neglects the presence of positive emotions which accompany the tingling sensation. It may be that ASMR is the positive end of a spectrum of a sound/emotion synaesthesia, and that this tingling sensation is a secondary phenomenon resulting from intensely positive feelings, rather than the primary concurrent. The data collected seems to support this, as many participants reported feeling relaxation and positive emotions even in the absence of a tingling sensation.

However, there is no mention in misophonia research of any negative counterpart to the tingling sensation found in ASMR. If one were looking for a truly polar opposite sensation, it may be expected to observe numbness in the skin or an irritating sensation present. It must be considered, however, that perhaps the opposite of this tingling sensation is not irritation, but actually the general level of sensation that might be expected in typical individuals. Rather than this aspect lying on a continuum from irritation to typical sensation to pleasant tingling, it is a smaller continuum between typical sensation and pleasant tingling, with many shades of grey between the two.

## Future directions for research

Though the age of the sample in the present study suggests engaging in ASMR is primarily an endeavour of young adults: this is likely to be reflective of limitations in the sampling method. Several individuals above age 40 provided input via this questionnaire, and some

participants spontaneously reported being aware that one of their parents and/or their children also experienced what they believe to be the same the sensation. This would suggest that the young age of the sample is more likely a product of user demographic of Facebook and Reddit than an accurate representation of ASMR capable individuals' ages. In the current study, synaesthesia was tested for consistency via e-mail interview. As the subtypes reported by participants were so varied, some immeasurable by the standard Test of Genuineness (TOG-R; *Asher et al., 2006*), interview was favoured over electronic tests of consistency such as those found on synaesthete.org (*Eagleman et al., 2007*). We suggest that future studies into ASMR include rigorous controls for synaesthetic experience.

While ASMR appears to be a genuine, relatively prevalent perceptual experience, the exact nature of the phenomenon is still unknown. There is the possibility that the tingles associated with ASMR result from a minor seizure, brought on by appropriate stimuli. This has been hypothesised in the past (*Novella, 2012*), but as of yet remains uninvestigated. In this vein, research utilising neuroimaging methods such as fMRI may further our understanding brain regions involved in ASMR. fMRI investigations in particular, however, have potential to prove problematic, as results of the current study show that individuals overwhelmingly require specific, quiet and relaxed conditions to achieve the desired sensation. An alternative avenue of research might be the use of so-called non-invasive brain stimulation (*Davis & van Koningsbruggen, 2013*) to modulate brain activity during ASMR. Techniques such as transcranial direct current or magnetic stimulation (tDCS, TMS) are known to induce multisensory experiences, often as an unintended side-effect of stimulation (*Davis et al., 2013*). Given the age demographic of ASMR consumers, we note that brain stimulation techniques should be used sparingly in younger people (*Davis, 2014*).

Further exploration into ASMR's relationship with Misophonia may also yield interesting results. Studies examining the co-occurrence of Misophonia and ASMR may shed more light on the possibility that these two experiences are related, or potentially opposite poles of the same spectrum. Similarly, the relationship with synaesthesia suggested by the results of this research should be taken further, using more robust consistency measures to verify that the high instance of synaesthesia in ASMR capable individuals. The suggestion that ASMR and Misophonia may be related was based primarily on similarities in reaction to auditory stimuli. Though sounds play a pivotal role in ASMR, it would also be advantageous to investigate the role of visual stimuli alone in triggering viewers, as such stimuli involving precise movements and focused tasks appear to be effective while being near silent.

## Conclusions

We have provided the first investigation into the phenomenon of autonomic sensory meridian response (ASMR). ASMR can be induced, in those who are susceptible, by a fairly consistent set of triggers. Given the reported benefits of ASMR in improving mood and pain symptoms, we suggest that ASMR warrants further investigation as a potential therapeutic measure similar to that of meditation and mindfulness.

### Funding

The authors declare there was no funding for this work.

### Competing Interests

The authors declare there are no competing interests.

### Author Contributions

- Emma L. Barratt conceived and designed the experiments, performed the experiments, analyzed the data, contributed reagents/materials/analysis tools, wrote the paper, prepared figures and/or tables, reviewed drafts of the paper.
- Nick J. Davis analyzed the data, contributed reagents/materials/analysis tools, wrote the paper, prepared figures and/or tables, reviewed drafts of the paper.

### Human Ethics

The following information was supplied relating to ethical approvals (i.e., approving body and any reference numbers):

Swansea University Department of Psychology Ethics Committee. Approval received via electronic statement: "Your proposed study 'An investigation into Autonomous Sensory Meridian Response as self-medication,' has been reviewed and is approved. Provided that the information obtained is kept absolutely confidential and that no personally identifiable information is entered on computer, you may proceed with your studies."

### Supplemental Information

Supplemental information for this article can be found online at http://dx.doi.org/10.7717/peerj.851#supplemental-information.

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
