# Peer review of "Autonomous Sensory Meridian Response (ASMR): a flow-like mental state"

_PeerJ, doi:10.7717/peerj.851_

## Round 0.1 · original submission · Minor Revisions

Sorry for the long review process. It took me quite a while to get reviewers. But now we have to positive reviews, #2 with minor points, which I would like you to handle accordingly.

I also tried to get a specialist in the field of migraine aura to review this article. None was available within a reasonable short time. You may want to look in future investigations into this condition, which in about 5% of all migraine cases causes no headache, but interesting mental states associated also with synesthesia (among many other phenomena).

·

Basic reporting

Basic reporting is fine

I think the introduction would benefit from discussion of the process by which a "lay phenomenon" is incorporated into the scientific lexicon. What does it mean, in effect, to ask "Is ASMR real?"? As psychologists we want to know if the people using the term are describing the same thing (or things), but we also want to know if they are describing some thing (or things) which already have a technical term. The aims of the study (as expressed in the final paragraph of the introduction) are broad. Discussion of the way a phenomenon is brought under scientific scrutiny might help justify why this has to be the case.

Lots of work on synaesthesia has been done since Ramachandran and Hubbard, 2001. I think the comparison of ASMR and synaesthesia is very instructive, so maybe the introduction would benefit from some additional consideration of this literature


Lines 143- 145 It is not clear if ppts rated their mood at these specific points, or if they rated their estimations of their mood at these hypothetical points (ie they provided all answers at the same time). I assume it is the latter, this could be made clearer

Experimental design

Experimental design is good

Validity of the findings

Figure 3 - this density plot looks nothing like a positive correlation of 0.96. Explain please

You might want to check correlation of BDI against change in mood (e.g. after-before). Looks highly significant to me.

Relatedly, can we have the data in non-proprietry format (eg .csv rather than .xlxs) please

line 283 "Synaesthesia appeared to be particularly prevalent within the sample"
- 5.9% vs 4.4% doesn't seem high to me. Isn't there a test you could do? Null H: prevalence is 4.4%. n = 475, what is p>5.9%? I bet you'd expect a higher than 0.05 proportion of samples with greater than 5.9% synaesthesia in a sample of 475 even with true rate of 4.4%

Additional comments

An important initial survey of an little researched area, well motivated, thoroughly conducted, clearly presented and highlighting the rich possibilities for future work.

Reviewer 2 ·

Basic reporting

This is an interesting paper reporting a survey on a topic, which has received lots of attention worldwide. It is about ASMR a phenomenon widely used and reported but with no substantial scientific support. Thus, this paper is urgently needed to start a scientific discussion about this phenomenon. In the following I have listed my comments and suggestions.

Introduction
The introduction is fine and covers many interesting aspects, which might be associated with ASMR. Some citations seem to be placed not correctly. For example, the Ramachandran & Hubbard citation refers to a grapheme-color synesthesia issue and not to the link between ASMR and synesthesia. But anyway, that is not that problematic.

At the end of the introduction the authors should describe that they plan to do a survey and that they plan to obtain subjective rating data. They should also why they used this line of research and not for example a neuroscientific or cognitive science approach utilizing more strongly experimental techniques.

Methods
Subjects: Is there some in addition available about the subjects? Profession or specific expertise? What about psychiatric diseases?

Whether the number of synesthetes in this group is larger as compared to a non-biased sample is unclear since the authors did not perform statistical tests. Anyway, I am pretty sure that there is no substantial difference to the unbiased sample.

Discussion
The discussion is fine. I am a bit skeptical whether there should be a special link to synesthesia. First of all, the number do not support this link (5.9 vs 4.4%). Second, synesthesia is a fast and automatic double perception which is simply there and is not supported by some kind of relaxation-like prerequisites.
Beside of that the discussion is fine and helpful to understan that problem.

Experimental design

This is a survey study. They have used state-of-the-art techniques.

Validity of the findings

The findings are fine based on the methods used.

Additional comments

Fine and important work

---

## Round 0.2 · accepted · Accept

Thank you for revising the manuscript.